# The challenge of phasing-out fossil fuel finance in the banking sector

J. Rickman [1,4], M. Falkenberg [2,4] ✉, S. Kothari [1], F. Larosa [3], M. Grubb [1] & N. Ameli [1] ✉

A timely and well-managed phase-out of bank lending to the fossil fuel sector is critical if Paris climate targets are to remain within reach. Using a systems lens to explore over $7 trillion of syndicated fossil fuel debt, we show that syndicated debt markets are resilient to uncoordinated phase-out scenarios without regulatory limits on banks' fossil fuel lending. However, with regulation in place, a tipping point emerges as banks sequentially exit the sector and phase-out becomes efficient. The timing of this tipping point depends critically on the stringency of regulatory rules. It is reached sooner in scenarios where systemically important banks lead the phase-out and is delayed without regional coordination, particularly between US, Canadian and Japanese banks.

The Paris Agreement goal is at risk without a rapid decline in fossil fuel consumption[1–3]. Projected emissions from existing fossil fuel assets could push the world past 2 °C of warming within decades[4,5]. This means investment in new fossil fuel assets is incompatible with agreed climate targets, and the financing of existing assets must be phased-out in line with well-defined decarbonisation pathways for a just and orderly transition. As banks are the primary source of financing for fossil fuel companies[6], their role in sustaining the 'business-as-usual' fossil fuel economy is now central to the wider climate debate[7–9].

Recently, the banking sector has attempted to align its activities with climate goals through voluntary actions. For instance, the UN-convened Net Zero Banking Alliance (NZBA) was launched in 2021 as banks came under increased pressure to decarbonise their lending portfolios in line with Article 2.1(c) of the Paris Agreement. However, signatories to the NZBA are being accused of greenwashing by NGOs and civil society, citing inadequate policy timelines and decision-making mismatched with net-zero rhetoric[7,8]. While the risk of fossil fuel assets becoming 'stranded'[10], i.e., devalued through concerted policy action, is perceived as a long-term risk[11–13], banks still have a financial incentive to hold short/medium-term investments in fossil fuel assets, largely due to their ongoing exposures to the sector.

The contribution of an individual bank to fossil fuel lending goes beyond its own direct investments through its participation in syndicated deals in which multiple banks pool their resources. Debt syndication is an important facility for the capital-intensive fossil fuel sector as it allows deals to be made that are too large for the balance sheets of an individual bank and spreads risk among syndicate participants. In this way, syndication amplifies the impact of individual banks' investments and drives significant financial support to the fossil fuel industry. Syndicated loans accounted for 66% of global fossil fuel finance in 2018, followed by bonds (29%) and equity instruments (5%)[6]. Indeed, bank loans may become ever more critical to fossil fuel firms in the future as capital markets are pricing climate-related financial risk[14,15], limiting the possibility for bank loans to be substituted with market-based finance[16]. Syndication also creates networks of lending relationships that facilitate the mobilisation of finance across the global banking system, reducing costly informational friction between borrowers and new lenders as new banks in a syndicate can benefit from the experience of existing banks[17,18]. In this way syndication facilitates the substitution of finance between banks exposed unevenly to climate policy[16,19,20] i.e., finance phased-out by domestic banks in countries with stronger climate policies can be substituted by finance from foreign lenders. This has been reported in Australia's coal lending market, for example, as Australian banks have pulled out of large loan syndicates to be replaced by Chinese and Japanese lenders[21].

Initiating a timely, ordered phase-out of fossil fuel finance requires an understanding of the syndicated fossil fuel debt market and its response to policy and regulatory measures. The behaviour of these markets, in turn, will depend on the underlying syndicated lending network, which is the focus of our study. In analysing the

[1]Institute for Sustainable Resources, University College London, London, UK. [2]Department of Mathematics, City University of London, London, UK. [3]KTH Royal Institute of Technology, Stockholm, Sweden. [4]These authors contributed equally: J. Rickman, M. Falkenberg. ✉e-mail: max.falkenberg@protonmail.com; n.ameli@ucl.ac.uk

structure and topology of the syndicated lending network, we add to the literature on financial networks[22–25] which has been key to understanding a wide range of phenomena relevant to financiers, governments, and regulators[26], as well as to the complex systems and network literature studying mechanistic processes related to socio-economic aspects of climate change[27,28]. In isolating just one module of the finance ecosystem—namely the network of syndicated lending relationships, which determines who lends with whom—our study is distinct from the stream of literature on financial stability, which describes shock transmission through networks of interbank liabilities[29–34]. Our results could feed into the related stress-testing tools and macroeconomic models[35–39] to provide a more detailed and realistic representation of the banking sector and its transition dynamics, which will be necessary for the coordination of an ordered transition.

Our analysis first describes the structure of syndicated lending networks[40–43], tracking the position of the critical lenders within this network since the Paris Agreement. It elucidates the dual role of banks as both direct lenders and syndicate participants, while highlighting the inconsistent behaviour of critical banks, particularly those in the US. It then investigates the phenomenon of finance substitution between banks to the fossil fuel sector and the potential role of regulation to limit such dynamics. As market failures and a lack of international coordination undermine the efficacy of market-based measures such as carbon-pricing instruments[44], central banks, and supervisors are indeed considering the regulatory instruments at their disposal to reallocate bank finance[45] due to the risks posed by climate change to financial stability[46,47]. Using a simplified network model of syndicated lending, our analysis illustrates how finance substitution can be counteracted by limiting banks' holdings and activities in the fossil fuel sector. This could be implemented, for example, by extending the Basel III framework as part of the existing work programme to incorporate climate-related financial risks across its three pillars. We further test a range of phase-out scenarios exploring the role of systematically important banks and the effect of regional coordination (or lack thereof). Our analysis underscores the significance of syndicated bank lending networks in shaping policy responses, especially as central banks and regulators advance their understanding of climate-focused regulatory frameworks.

## Results

### Fossil fuel lending has shown no systematic decline over the last decade

To analyse syndicated fossil fuel lending, this study uses data from Bloomberg, which reports syndicated debt (loans and bonds) provided by banks to fossil fuel companies between 2010 and 2021. The data reports $7.1 trillion of bonds and loans extended by 709 banks, the majority of which are syndicated (81% by volume, Fig. 1a). Bloomberg data provides industry-leading coverage of the global banking sector, with reported global debt volumes surpassing those of alternative financial data providers by 29% (see SI Methods). Nonetheless, the data lacks information on fossil fuel borrowers, such as their location or asset type (coal, oil, or gas). While these dimensions of the fossil finance landscape are relevant for just transition planning at the country-level, we focus our analysis on the dynamics of global fossil fuel debt supply, in support of the efforts of international financial supervisors.

Dynamics in the syndicated fossil fuel debt market are distinct from the global syndicated loan market as it experiences sector-specific phases of growth and contraction related to economic, geopolitical, and technological factors (Fig. S1). Fossil fuel lending has shown no systematic decline over the last decade (Fig. 1a). Banks provided $592 billion of bonds and loans for oil, gas, and coal companies in 2021, compared to a yearly average of $584 billion between 2010 and 2016 (the year the Paris Agreement was signed). The

distribution of banks' lending volumes is, however, highly skewed (Fig. S2); the top 30 banks provided 78% of total lending between 2010 and 2021. JP Morgan, for example, the largest lender, had an 8% market share in 2021 while the median market share of fossil fuel lenders was just 0.03%.

Phasing-out fossil finance provided by the top tier of lenders would significantly limit the amount of finance available for substitution, due to their size and market dominance. In order to assess the progress of their phase-out, we compared the average new exposure (new fossil fuel assets as a percentage of their total assets) of the top 30 lenders (Fig. 1b) and their average annual lending between the pre-Paris Agreement (2010–2016) and post-Paris Agreement (2017–2021) period (Fig. 1c). A select cohort of European banks have led the way in the fossil finance phase-out; the four largest banks to decrease their yearly lending by more than 40% from pre- to post-Paris are Swiss (UBS, Credit Suisse), German (Deutsche Bank) and Norwegian (DNB ASA) (Fig. 1c), likely reflecting the stronger climate commitments and policy stringency in the region prompting European banks to price transition risk higher[48]. However, these decreases appear to be offset by increased lending from large banks in other regions with weaker climate policies. In particular, two Canadian banks (Scotiabank and BMO Capital Markets) and three Japanese banks (Sumitomo, Mitsubishi UFJ and Mizuho) made substantial increases to their average annual fossil fuel lending (>25% from pre- to post-Paris, Fig. 1c). In many cases this increase in fossil fuel lending corresponds to a sector-specific, rather than bank level, change in activity reflecting the strategic priorities of these banks (Fig. 1b). The 'big-four' US banks (JP Morgan, Citi, Wells Fargo, and Bank of America) continue to dominate the market and made relatively small decreases in their lending from pre- to post-Paris (Fig. 1c). Bank of America decreased its lending by 25% on average from the pre- to post-Paris period, while the others decreased their lending by less than 10% (Fig. 1c).

### Syndicated lending networks determine the systemic importance of banks

To measure the influence and potential impact of banks in the syndicated fossil fuel lending network, we define banks' 'syndication activity' as the total value of deals in which a bank is a syndicate partner, expressed as a percentage of all deals made in a given year. Syndication activity thus reflects the wider impact of an individual bank's phase out, as all deals it was party to must secure substitute finance.

In the syndicated lending network links are made between banks, weighted according to the number of times that they have invested together in a fossil fuel debt syndicate—reflecting the strength of their lending relationship. Figure 2a shows that the systemically important banks, with high syndication activity, are highly interconnected amongst themselves and sit at the core of the lending network. These banks with high network centrality also act as bridges to banks that sit at the network periphery (Table S1, Methods). Through their connections, the core banks are thus able to draw in finance from peripheral banks as well as pooling their own finance.

The composition of the network core has changed over time (Fig. 2a). Several European banks have made a retreat from the core; Deutsche Bank, for example, goes from the 6th most centrally positioned bank to the 19th. While Japanese and Canadian lenders moved into the core; Mizuho Financial, for example, increased its centrality ranking from 9th to 5th.

None of the big-4 US banks increased their individual lending between the pre- and post-Paris period, however they have retained their positions in the network core and have increased their syndication activity (Fig. 2a). Citi's average annual lending, for example, decreased from $35 bn pre-Paris to $34 bn post-Paris period but its syndication activity increased from 32% to 37% (Fig. 2a). Banks can, in this way, increase their influence in the sector while at the same time lowering their direct investments.

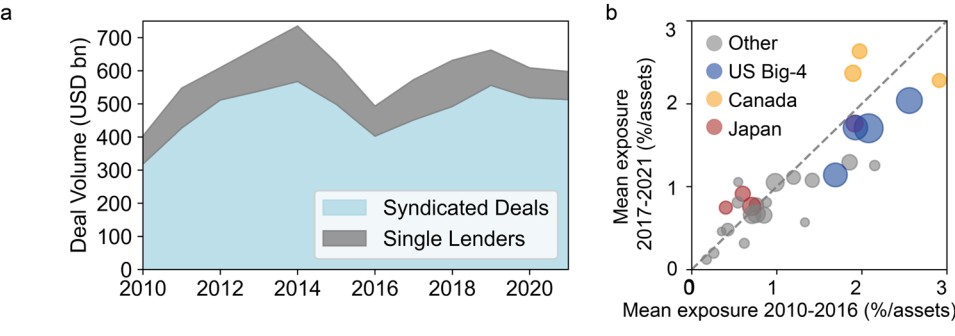

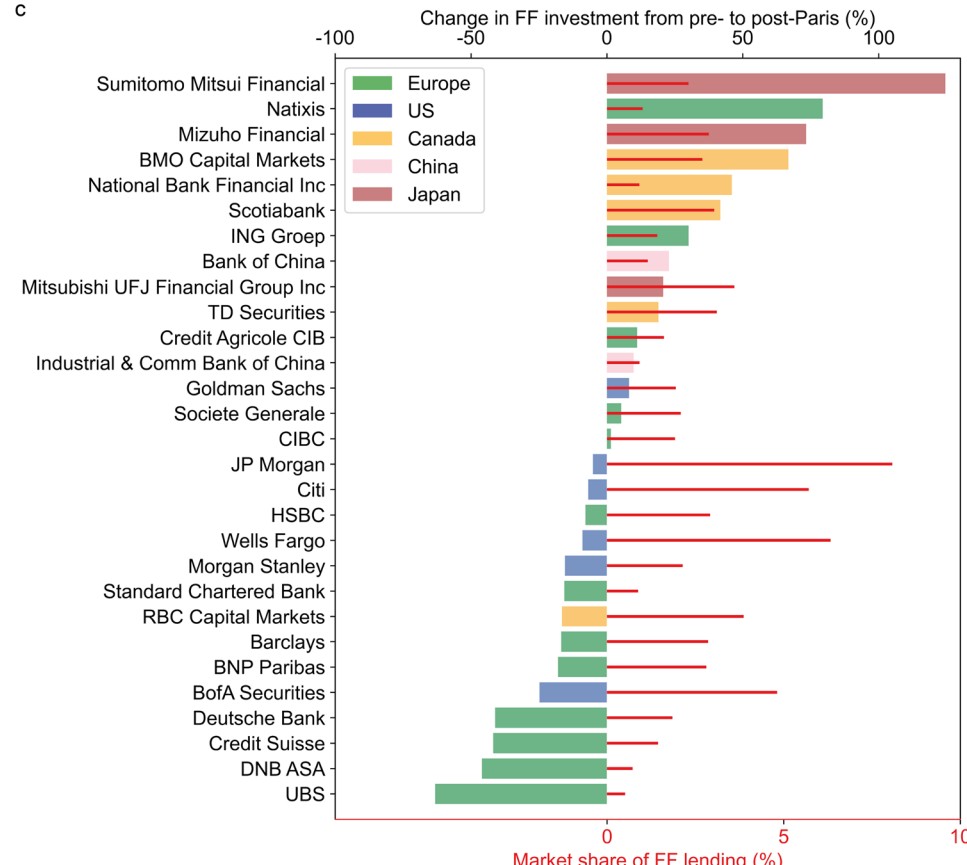

**Fig. 1 | Lending activity of the banking sector to fossil fuel companies. a** Debt provided by banks to the fossil fuel (FF) sector between 2010 and 2021, separated into syndicated deals and deals with a single lender. **b** Mean new fossil fuel exposure for the top 30 banks in the pre-Paris Agreement (2010–2016) and post-Paris Agreement (2017–2021) period. **c** Bars (coloured according to the location of banks' headquarters) show the percentage change in the average annual lending of the top 30 banks between the pre-Paris Agreement and post-Paris Agreement period. Red lines show banks' market share, defined as their individual lending in the post-Paris period as a percentage of total market lending.

## Modelling syndicated lending and bank phase-out

The changing composition of the network core points to a substitution effect, whereby banks pulling out of lending syndicates are replaced by others. As an example, Fig. 2b plots the share of Citi's co-investments made with key syndicate partners over time (the number of edges between Citi and a given partner bank in the co-investment network, as a percentage of Citi's degree). Deutsche Bank and UBS have pulled out of syndicates involving Citi, but their finance is substituted by other banks such as Sumitomo Mitsui and Mizuho Bank. This phenomenon of finance substitution shows how finance flows to fossil fuel companies can be resilient to the phase-out of individual banks[49]. To investigate this further, and to explore the role that prudential regulation could play in countering this phenomenon, we introduce an illustrative network model that simulates banks phase-out from the sector

and the substitution of their phased-out finance by other banks. With this model our aim is to elucidate the challenge posed by finance substitution within the syndicated lending network and provide preliminary evidence to inform the modelling work of central banks and regulators.

The model, described in detail in the Methods, considers how syndicated fossil fuel deals acquire substitute finance as banks phase-out from the fossil fuel sector. In the basic formulation of the model, banks sequentially phase-out from the sector removing all of their finance from the deals in which they are syndicate partners. Then, deals which are affected by the phase-out—the 'deals-at-risk'—attempt to find new syndicate partners to provide substitute finance. If these deals acquire substitute finance, the phase-out is *inefficient* as bank-level phase-out does not result in fewer fossil fuel deals being financed.

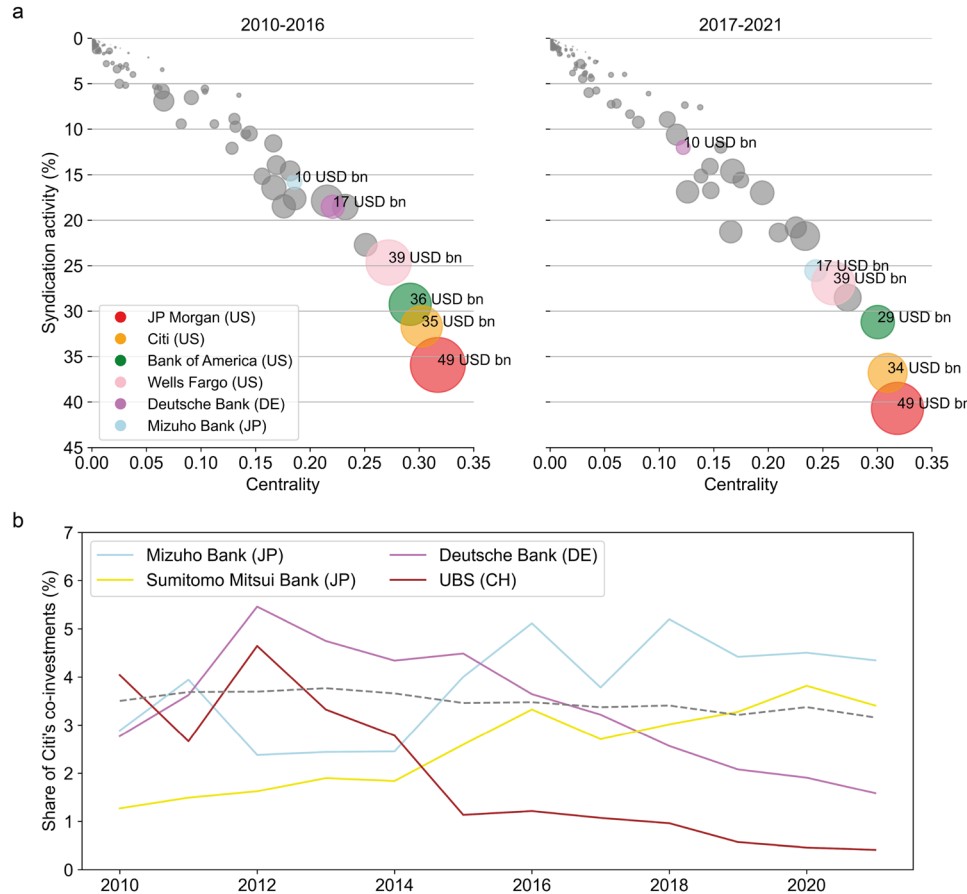

**Fig. 2 | The topology of the syndicated lending network and the substitution effect. a** Banks' eigenvector centrality in the syndicated lending network is plotted against their average syndication activity in the pre-Paris Agreement (2010–2016) and post-Paris Agreement (2017–2021) period. Pearson's correlation coefficient between network centrality and syndication activity is 0.98 in each period. The size of each node represents banks' average yearly lending. Selected banks are coloured and annotated with their average yearly lending. **b** Time series plot shows changes in the co-investment activity between Citi and its syndicate partners. Selected banks (coloured lines) increased or decreased their share of Citi's co-investments by greater than 1 percentage point between the pre-Paris and post-Paris period. The average share of Citi's top 20 syndicate partners is shown by the dashed grey line.

Conversely, if the deals-at-risk do not secure substitute finance, they fail, and the phase-out is considered *efficient*. To model prudential regulations that could restrict finance substitution, we impose an upper limit on the value of new fossil fuel finance that a bank can take on its books in a given year by setting a 'finance limit percentage'. This percentage, defined as the maximum annual percentage increase in a bank's fossil fuel lending (see Methods), serves as a generic analogue for prudential measures limiting banks' involvement in the fossil fuel sector. For example, if Bank A held fossil fuel assets worth $1bn in 2021, a 10% finance limit percentage would set a cap on Bank A's 2022 fossil fuel assets at $1.1bn. Data shows that year-on-year changes in banks' fossil fuel exposure typically fall in the range [-60%, +160%] (95% CI; Fig. S3). We therefore choose a feasible range of the finance limit percentage to be between a 0% and 200% limit, with smaller finance limit percentages representing more stringent regulation. Model results are qualitatively robust to setting the cap as a limit to fossil fuel exposure (e.g., new lending cannot exceed 1% of total assets, Fig. S4). However, given significant variability in exposure across the largest banks (Fig.1b), such flat-rate limits only affect the most exposed banks (notably in Canada and the USA) unless the exposure limit is very strict.

Using this base model, we test how restrictions to finance substitution impacts the efficiency of banks' phase-out from the fossil fuel sector under different scenarios (e.g., random, targeted, and regional phase-out scenarios). In all scenarios, the structure of the syndicated lending network and the value of fossil fuel deals is taken directly from the Bloomberg data. The structure of the syndicated lending network

considers banks which are syndicate partners in fossil fuel deals only, and it does not consider banking relationships in other sectors. For simplicity, the results show data from syndicated deals made in 2021. Qualitative results for the base model are robust across all years in the dataset (see Fig. S6 for other years).

The model makes a number of simplifying assumptions: (i) bank 'phase-out' is conceptualised as the complete cessation of fossil fuel investment and does not distinguish between asset types (i.e., coal, oil, or gas), (ii) banks phase-out from the sector sequentially, (iii) finance can be substituted only by a bank not already a syndicate partner in a given deal, and (iv) prudential regulation imposes a strict limit on banks' fossil fuel financing while additional behavioural adaptations such as recapitalisation are not considered. These assumptions are justified based on the illustrative nature of our model which could be developed in more detailed and sophisticated directions by, for example, gaining access to banks' supervisory data.

## Financial regulation can weaken syndicated fossil fuel lending networks

In the base version of the model, banks phase-out from the fossil fuel sector in a random order and any remaining bank, selected randomly, may provide substitute finance to deals-at-risk (referred to as the 'any substitute' scenario), provided they do not exceed the cap set by the finance limit percentage (see Methods).

We define a phase-out 'efficiency' by comparing the value of failed deals to the value of deals-at-risk (see Methods). In a maximally efficient

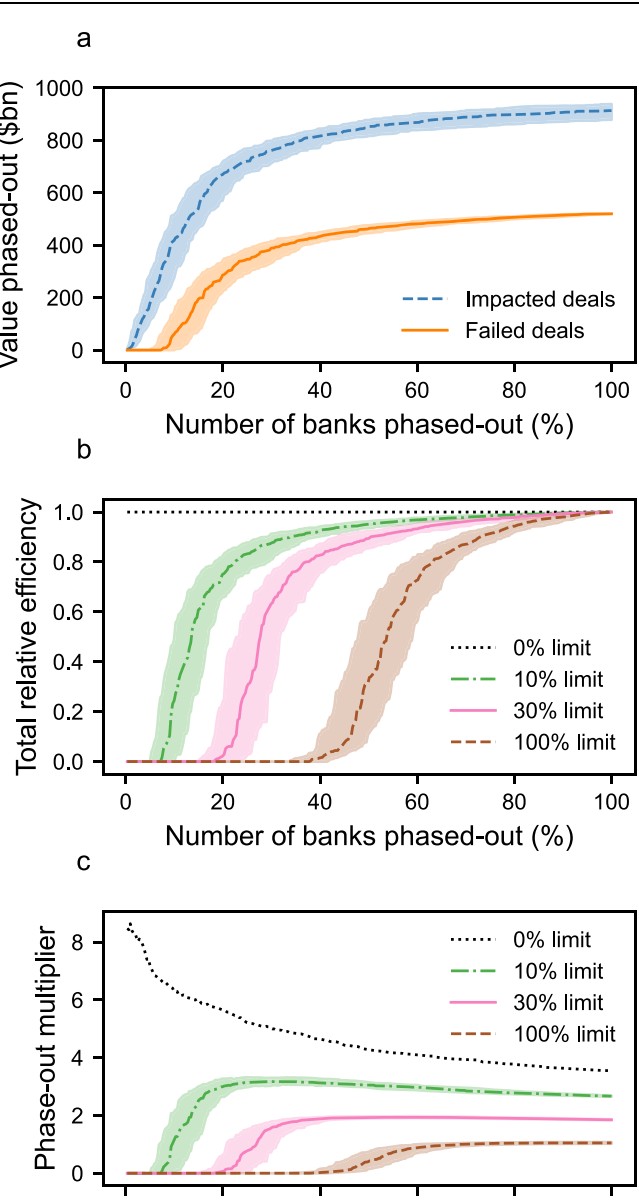

**Fig. 3 | Phase-out value, efficiency, and multiplier for different finance limit percentages. a** Banks exit the fossil fuel sector in a random order (321 banks total). The cumulative value of impacted deals using a finance limit of 10% (blue dashed); this includes both the deals which do, and do not, succeed in acquiring substituted finance. The cumulative value of the deals which fail (orange solid). **b** The total relative phase-out efficiency (see methods for definition) for different finance limits in the phase out model. If few banks exit the sector, efficiency is low since deals successfully substitute finance. Beyond a tipping point, efficiency grows rapidly. **c** The total phase-out multiplier (see methods for definition) for different finance limits. A multiplier larger than 1 indicates that the direct phase-out of finance by one bank has forced the indirect phase-out of finance from syndicate partners. All curves correspond to the median value across 100 random simulations. Shaded regions correspond to the inter-quartile range across the 100 simulations.

phase-out, all deals-at-risk fail because they cannot secure substitute finance resulting in a phase-out efficiency score of 1. However, if some deals-at-risk find substitute finance and survive then phase-out efficiency is less than 1 (becoming zero if all deals-at-risk successfully acquire substitute finance). This results in an 'efficiency gap' between efficient and inefficient phase-outs, which is expressed as a multiple of the total value of fossil fuel deals (see Methods and Fig. 3a, b).

Figure 3b shows the phase-out efficiency for different finance limit percentage values. Initially, when only a few banks have exited the sector, many banks can step in to provide substitute finance, resulting in near-zero phase-out efficiency. As more banks exit the sector, there comes a tipping point where potential substitute banks reach their finance limit, causing phase-out efficiency to sharply rise. The number of banks needed to phase-out before reaching the tipping point decreases as the finance limit percentage becomes stricter; hence the tipping point is reached sooner under more stringent regulations (Fig. 3b).

Figure 3c shows the 'phase-out multiplier', which is the ratio of the finance removed directly by a phasing-out bank to the value of deals failing due to its phase-out (see Methods). When substitution is possible, the phase-out multiplier is zero before reaching the tipping point, since all phased-out finance can be replaced. Beyond the tipping point two types of substitution dynamics emerge. With sufficiently large finance limit percentages (>100%) the phase-out multiplier never exceeds 1. This means that, on average, $1 removed by the phased-out bank results in less than $1 of finance being phased-out from the sector. Conversely, with sufficiently small finance limit percentages (<100%), the multiplier exceeds 1. This indicates that $1 removed by the phasing-out bank results in more than $1 of finance lost to the sector. This occurs because the direct phase-out of an individual bank induces indirect phase-out of its syndicate partners as their finance is withdrawn from failed deals, while they remain active in the sector. Finally, when the finance limit percentage is 0%, no substitution is possible, and phase-out efficiency remains at 1 throughout the phase-out process. In this case, every phasing-out bank indirectly induces the phase-out of other banks, resulting in a large phase-out multiplier. This effect is strongest at the start of the phase-out process when the average number of deals per active bank is highest (since a single bank's phase-out results in many failed deals). As the phase-out continues, the average number of deals per active bank decreases (as many deals have already failed), and the indirect phase-out effect weakens, leading to a gradually decreasing phase-out multiplier.

Results remain consistent when using syndicate data from other years (2010–2020), with minor changes in the efficiency gap over time (Fig. S6 and SI Methods). Analysis of yearly data from 2010 to 2021 reveals that it has become gradually harder to reach the tipping point from inefficient to efficient phase-out over time, given a fixed finance limit percentage. This suggests that increased syndication activity by top lenders (Fig. 2a) has strengthened the resilience of the syndicated fossil-fuel lending network to unregulated phase-out scenarios.

## Fossil fuel phase-out is optimised by targeting systemically important banks

We now consider a second 'targeted' phase-out scenario, where the most influential banks in the sector exit first. We compare this approach to the random phase-out scenario (Fig. 3). For both cases, we use the 'any substitute' rule for determining candidates for substitution (any bank may provide substitute finance). The targeted scenario is motivated by post-2008 financial crash policy discussions aimed at regulating systemically important 'too-big-to-fail' financial institutions[50]; in our context these are the institutions providing the largest volume of finance to the fossil fuel sector. As shown in Fig. 4a, an efficient phase-out requires significantly fewer banks to leave the sector in the targeted scenario compared to the random scenario (see also Fig. 4d), since the largest banks contribute disproportionately to overall fossil fuel finance. Although the tipping point from inefficient to efficient phase-out occurs at comparable values of withdrawn finance in both scenarios (Fig. 4b), the targeted scenario does result in an important increase in phase-out efficiency, presented as a decrease in the efficiency gap (Fig. 4c). This is because once the largest banks have phased-out in the targeted scenario, unlike in the

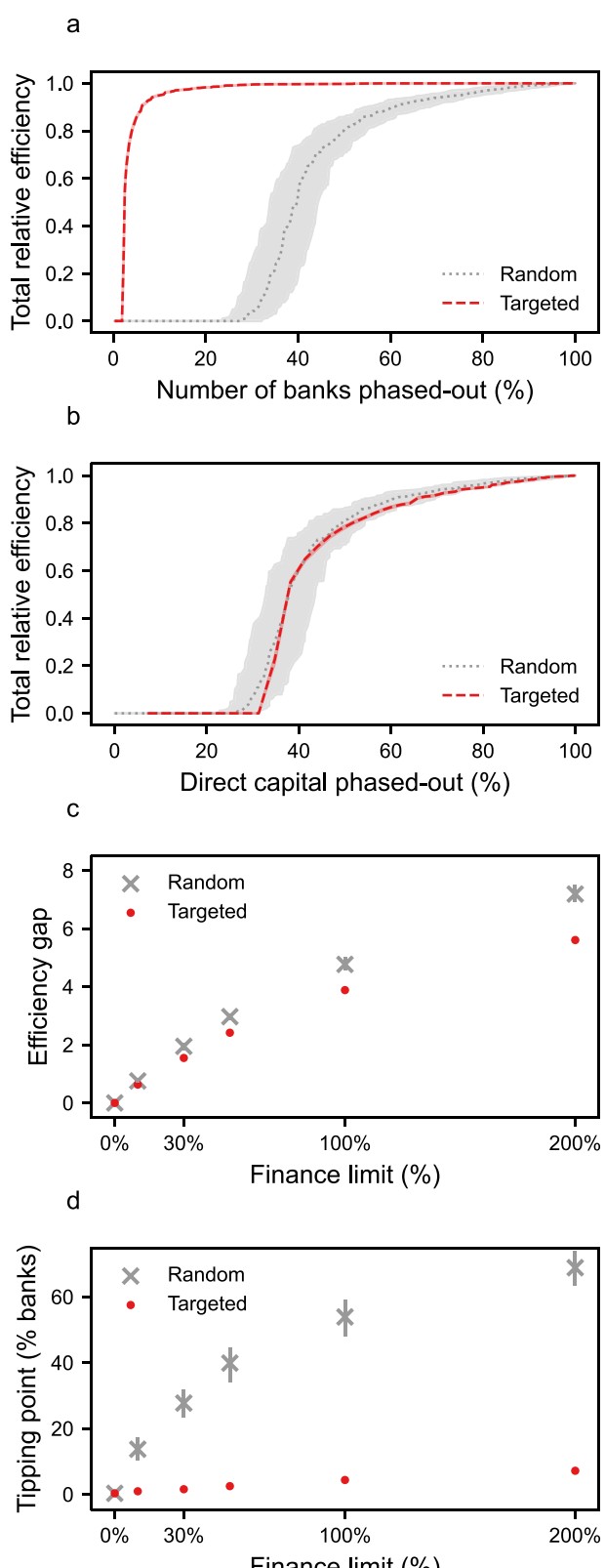

**Fig. 4 | Phasing-out banks in order of importance increases phase-out efficiency. a** Total relative efficiency as a function of the percentage of banks which have phased-out from the fossil fuel sector. Targeted removal of banks (red dashed) reduces the number of banks which must be phased-out to achieve non-zero efficiency, relative to the random case (grey dotted). For illustrative purposes, both scenarios use a finance limit of 100%. **b** The equivalent to **a**, but as a function of the total capital phased out directly by banks. **c** The efficiency gap for various different finance limit percentages. For a fixed limit, the efficiency gap is smaller for targeted phase-out (red points) than random phase-out (grey crosses). **d** The tipping point, defined as the percentage of phased-out banks required to reach a total relative efficiency of 0.5. All panels show the median value from 100 simulations for each model. Shaded regions correspond to the inter-quartile range across the 100 simulations.

scenario we examine a scenario where potential substitutes are identified based on the structure of the syndicate network itself (the 'syndicate substitute' scenario). In this case, banks eligible for substitution are chosen from the historical syndicate partners of banks who remain active in each deal-at-risk, thereby leveraging their existing relationships within the syndicate network to introduce new partners that can substitute phased-out capital. These candidate banks are ranked based on the frequency of their past collaboration with any current syndicate member. This ranking establishes the list of potential substitutes for finance, with the top-ranked bank chosen as the first candidate (see Methods).

Figure 5a shows the phase-out efficiency for the 'any substitute' and 'syndicate substitute' scenarios. For illustrative purposes, both scenarios use a finance limit percentage set at 100%, and banks are phased-out from the sector in a random order. The targeted phase-out scenario using syndicate substitution is shown in the supplement (Fig. S5). In the early stages of the phase-out process the 'any substitute' scenario has an efficiency of zero since all deals are able to secure substitute finance, whereas the 'syndicate substitute' scenario has a small non-zero efficiency. This is because deals financed by banks with weak connections to the core of the syndicate network, or by a single bank, fail to find substitute finance. As the phase-out progresses, both scenarios reach a tipping point where phase-out efficiency rapidly increases. However, this point is reached earlier in the 'any substitute' scenario than in the 'syndicate substitute' scenario. This discrepancy occurs due to an important network effect: in the 'syndicate substitute' scenario, substitute finance is more likely to originate from large, systemically important banks (see Fig. 2a). Since these banks have larger finance caps (in absolute terms) than smaller banks, the probability of successful substitution is higher, allowing deals to survive for longer into the phase-out process (see efficiency gap charts in Fig. S7).

In the 'syndicate substitute' scenario, any ranked bank may act as a candidate for finance substitution. We now consider more restrictive scenarios where only banks with the strongest links to the current syndicate are eligible for substitution, limiting the candidate pool to the top N ranked banks for each syndicate. Figure 5b shows this additional constraint and highlights how further limiting the potential candidates for capital substitution results in an earlier tipping point from inefficient to efficient phase-out. Finally, a deal may fail not only due to a lack of candidate banks for substitution, but also because it becomes perceived as excessively risky by investors after many attempts to secure substitute finance. To capture these dynamics, we implement a model where a deal may only acquire substitute capital a fixed number of times before failing. Figure 5c shows that limiting the number of times a deal can substitute finance results in an earlier tipping point.

Overall, all model variants show qualitatively consistent behaviour: when there are no restrictions on the substitution of bank finance in fossil fuel deals, the phase-out of finance from the sector is inefficient. However, when restrictions on substitution are introduced,

random scenario, the remaining banks are individually too small to provide substitute finance to deals-at-risk, even if there is sufficient available finance across the sector as a whole.

## The impact of additional restrictions on capital substitution
We now consider how the structure of the syndicate network may influence finance substitution. Contrasting the 'any substitute'

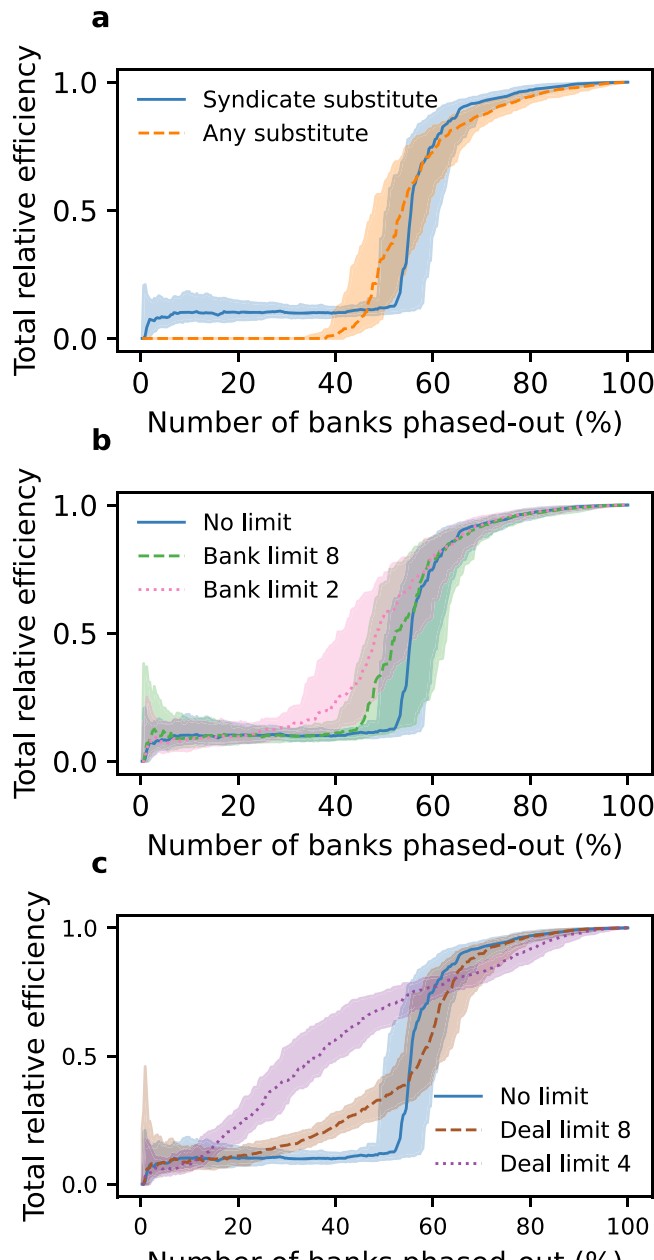

**Fig. 5 | Scenarios testing the effect of additional limits to finance substitution on phase-out efficiency.** For all panels, total relative efficiency is plotted as a function of the percentage of banks which have phased-out from the fossil fuel sector. Banks exit the sector in a random order (321 banks total). For illustrative purposes, all panels use a finance limit of 100%. **a** The total relative efficiency when any active bank can provide substitute capital to a deal at risk ("any substitute"), and when only active banks with a relationship to the current deal-at-risk's syndicate can act as substitute partners ("syndicate substitute"). **b** The syndicate substitute scenario, but where the number of candidate banks is limited to the top N banks which are co-active with the current deal-at-risk's syndicate. **c** The syndicate substitute scenario, but where the number of times a deal can acquire substitute capital is restricted, after which the deal must fail. The model variants shown in **a**–**c** are discussed in greater detail in the supplement (Figures S7–S9). All curves correspond to the median value across 100 random simulations. Shaded regions correspond to the inter-quartile range across the 100 simulations.

the phase-out process can become efficient once a tipping point is reached. This tipping point occurs earlier if the restrictions to substitution are more stringent.

## The impact of prudential regulation rules at a regional level

The results thus far assume that prudential regulations limiting bank lending to fossil fuel companies are implemented globally by, for example, extending the Basel III framework on capital requirements rules. We now consider scenarios in which banks exclusively phase-out from specific regions.

As of 2021, the countries whose banks finance more than 5% of the fossil fuel sector are the USA (33%), Canada (18%), Japan (11%), China (7%), France (7%) and the UK (7%). Banks within the European Economic Area collectively finance approximately 15% of the sector. Consequently, these regions are critical in our analysis of regional phase-out scenarios.

Under the any substitute rule (see Fig. 3), exclusively phasing-out banks from specific regions will be inefficient since banks with no prior relationships to the current syndicate could still provide substitute finance to deals-at-risk. As an alternative, we consider a regional phase-out scenario using the syndicate substitute rule, which captures the regional structure of banks' lending relationships. In this scenario, only banks within the affected region are phased out, while banks from other regions are not subject to regulation; thus, the finance limit percentage does not affect the results of this scenario. However, the banks external to the region under regulation must have lending relationships with banks within the region to provide substitute finance.

Figure 6a shows regional investment volumes successfully phased out when banks from specific regions exit the fossil fuel sector, plotted against each region's total investment volume. Substitute finance provided by banks external to the phasing-out region results in inefficient phase-out across all six regions analysed. However, there are substantial regional differences in phase-out efficiency. For the UK, Japan, and the EU, less than 20% of regional investment volumes are successfully phased out. This outcome is largely due to the large volume of deals acquiring substitute finance from US banks (Fig. 6b). In contrast, approximately a quarter of the US volume, half of the Canadian volume, and, most notably, around three-quarters of the Chinese market can be phased out successfully. The Chinese case presents a striking difference compared to the UK, despite both regions representing comparable shares of global lending volumes at 7% each. While UK banks have strong syndicate ties to banks from other regions, the Chinese sector is largely isolated from non-Chinese banks.

Figure 6b shows the regions from which substitute finance is most likely to be provided if banks from a specific region phase-out from the fossil fuel sector, revealing important interdependencies between regions. Firstly, when non-US regions undergo phase-outs, US banks emerge as the most likely candidates to provide substitute finance (as indicated by the arrows pointing towards the US node) illustrating their global importance. Second, US banks have strong, mutually supporting relationships with Japanese banks and Canadian banks. In contrast, while US, Japanese and Canadian banks are likely to substitute finance phased-out by EU banks (indicated by arrows from the EU to these countries), EU banks are unlikely to provide substitute finance if Canadian and Japanese banks phase-out (no arrows from Japan or Canada to EU). Additionally, EU banks play a minor role in providing substitute finance to US banks. This is consistent with the trends depicted in Fig. 2, illustrating the growing centrality of Japanese banks, and the receding centrality of EU banks. Finally, UK and Chinese banks primarily maintain meaningful relationships with US-based banks, but in the latter case the total size of capital flows is relatively small.

## Discussion

Our study explores the network of bank lending relationships that underlie the syndicated fossil fuel debt market. We aim to elucidate the network structure, the behaviour of critical actors, the phenomenon of

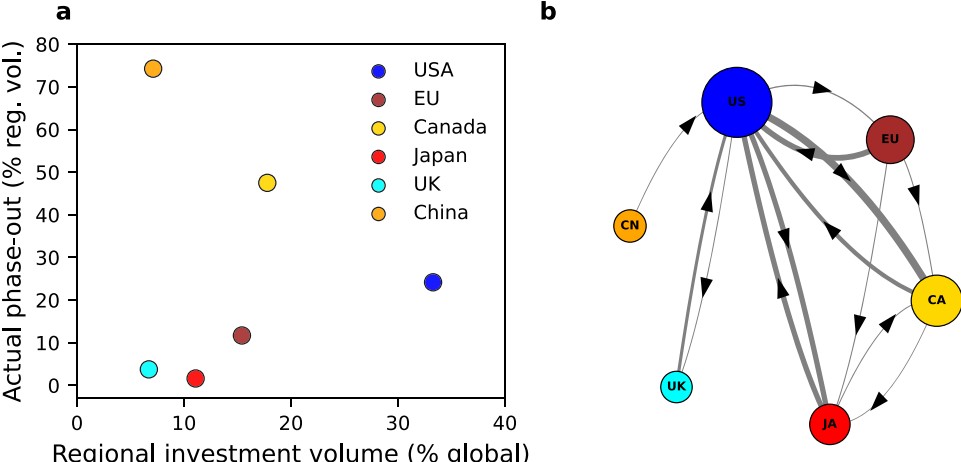

**Fig. 6 | Efficacy of prudential regulations at the regional level.** The phase-out model uses the syndicate substitution rule, in which only banks from a specific region are phased out. **a** The percentage of regional investment which is successfully phased-out from the fossil fuel sector, as a function of the region's total investment volume. **b** A network showing the regions from which substitute finance to deals-at-risk is most likely to be provided due to the phase-out of banks from a specific region. Node size is proportional to the regional investment volume. Edges between nodes indicate the flow of substitute finance (i.e., an arrow from China to the US indicates that US banks provide substitute finance to deals-at-risk when Chinese banks phase out), with the edge width proportional to the volume that is successfully substituted. Flows less than 1% of the global volume are omitted, and banks from other regions—totalling less than 10% of the global market—are ignored.

finance substitution between banks, and examine how the network responds to regulation that limits banks' lending to the fossil fuel sector. At an aggregate level there has been no decline in fossil fuel lending since the Paris Agreement, but regional differences are apparent. A select cohort of European banks have led the fossil finance phase-out by decreasing both their individual lending and their influence within the deal syndication network. Conversely, several large Japanese and Canadian banks have increased their individual lending and syndication activity—indicating a trend of finance substitution. While the 'big-4' US banks have made some progress in decreasing their overall exposure to the sector, their centrality in the deal syndication network continues to increase. This reflects their continuing influence in the sector, sustained by active lending relationships that span across regions. This dual aspect of banks' fossil fuel lending activity highlights their role both as direct financiers and as syndicate participants capable of enabling finance substitution.

The substitution of finance between banks in syndicated deals inhibits a sector-wide phase-out of new fossil fuel finance, posing a key challenge to the transition away from fossil fuels. We identify an 'efficiency gap' between the value of deals put at risk by banks phasing-out their finance and the value of deals that fail to secure substitute finance. We highlight the potential role of prudential regulation in counteracting this substitution phenomenon. We observe a sharp transition from inefficient to efficient phase-out as banks sequentially leave the sector and substitutable finance runs out due to regulated limits on banks' lending. The occurrence of this tipping point depends critically on the stringency of regulatory rules, with the number of banks phasing-out before the tipping point decreasing rapidly as prudential regulation is tightened. Moreover, under stringent regulation, the direct phase-out of one bank can result in the indirect phase-out of others, amplifying the overall impact.

In addition, the phase-out of fossil fuel finance is more efficient and requires fewer banks to exit the sector in a scenario targeting the exit of the largest lenders, compared to a scenario in which banks phase-out randomly. Influential banks should, therefore, strategically coordinate their transition plans to enhance their collective impact on syndicated debt markets. This is particularly crucial for the major lenders who facilitate the most deals and mobilise finance from smaller banks. Of the top 30 banks reviewed in our data, 23 are signatories to the NZBA. While such alliances have the potential to facilitate a rapid

and efficient phase-out of fossil fuel lending, the actual coordination between NZBA signatories and the achievement of significant phase-outs remains uncertain[8].

Regional regulation could further influence the success of prudential rules. Specifically, US banks, and to a lesser extent Japanese and Canadian banks, are key players sustaining the syndicated fossil fuel lending market due to their inter-connections with banks globally. In regional phase-out scenarios these banks could potentially provide substitute finance and delay reductions in fossil fuel flows. Consequently, regional regulation must account for the highly interconnected and international nature of syndicated fossil fuel lending. Chinese banks are a notable exception to this trend as the isolation of the Chinese banking sector means that the phase-out of Chinese banks could drive an important decline in the fossil fuel deals successfully financed.

A timely and ordered phase-out of fossil fuel finance will require action by financial regulators. Regulation to limit bank lending to the fossil fuel sector could take the form of capital requirements rules, for example, that are developed through formal standard-setting bodies such as the Basel Committee on Banking Supervision and networks of central banks and prudential regulators e.g., the Network for Greening the Financial System and the Financial Stability Board. Macroprudential instruments, such as capital requirements rules, can have direct and indirect impacts on the financial system and the wider economy, the full extent of which is not yet well understood. These impacts can vary depending on a country's economic reliance on fossil fuels[51] which necessitates just and managed transition planning[52]. Indeed, tipping point dynamics in the banking sector have the potential to spill over into other sectors leading to a cascade of nonlinear changes that could drive transformative shifts in the real economy[53,54].

Prudential institutions and central banks, tasked with maintaining financial stability, are calling for further research on such issues[55]. Our insights into the networks of lending relationships underlying syndicated fossil fuel debt could inform this work, particularly when combined with research on climate risk assessment and stress-testing methodologies. Furthermore, network dynamics identified here could be used to develop more detailed, bottom-up representations of the banking sector to be used in agent-based integrated assessment models assessing climate damages[56]. This would allow

for a more comprehensive understanding of potential financial vulnerabilities arising from climate-related risks within bank lending networks.

## Methods

### Data description

The data is sourced from a proprietary Bloomberg dataset which reports on debt (bonds and loans) provided to the fossil fuel sector by banks between 2010 and 2021. Fossil fuel companies are defined as companies involved in the operation, exploration, production, refining and marketing of coal, oil and gas assets. The dataset reports financing details of 14,391 bonds and loans totalling $7.1 trillion including the size of the deal and the bank(s) providing the finance; the majority of debt is provided as loans, which can be issued at the corporate level or as project finance. The remaining finance is provided as bonds. Comparing Bloomberg data with studies using alternative industry-leading sources of financial data[57] affirms that Bloomberg provides excellent coverage of the global banking sector (SI Methods).

### Syndicated lending network

The syndicated lending network is constructed by forming links between banks to represent lending relationships (i.e. co-investments). Links are weighted according to the number of times banks have been co-participants in a fossil fuel syndicate. Relationships based on syndicated deals in other sectors are not considered. A set of centrality measures[58] are used to assess the topology of lending networks and the positions of banks within it (Fig. 2a, Table S1): (i) Eigenvector centrality measures the influence of nodes in a network. Relative scores are assigned to each node such that a high-scoring node is connected to many nodes that themselves have high scores. (ii) Betweenness centrality measures a node's influence with respect to the flow of information in a network. Nodes with high betweenness centrality act as bridges from one part of the network to another. (iii) Closeness centrality measures a node's ability to efficiently spread information within a network. High closeness centrality of a node indicates short distances to other nodes. (iv) Degree centrality counts the sum of a node's connections to other nodes.

### Phase-out model

In the phase-out model, debt data are represented as a bipartite network between a set of N banks and M fossil fuel deals (Fig. 7). Bank $i$ providing finance to deal $j$ is represented as an edge between the bank and deal in the bipartite network. Edges are weighted according to the estimated loan or bond value provided to the deal by the bank such that the sum of all edge weights connected to a specific deal is equal to the total value of that deal. Note, the dataset only provides the total value of a deal at the syndicate level, and not at the bank level. Therefore, we assume that each bank in a syndicate provides an equal share of the finance to the deal in question.

The finance limit for each bank is set as the total lending of that bank (i.e., the sum of that bank's edge weights), plus a percentage factor which is a key model parameter, the 'finance limit percentage'. Importantly, note that if a bank is forced to phase-out of specific deals-at-risk indirectly—due to the phase-out of a partner bank and the subsequent failure of finance substitution—the finance which was assigned to the failing deal is not freed up for use in other deals-at-risk (i.e., finance phased-out indirectly cannot be reassigned to new deals). In the supplement, we show that the model is qualitatively robust if finance from failing deals is freed up for use in other deals-at-risk, although the reassignment of finance does weaken the efficacy of the finance limit and prevents the indirect phase-out of finance (Fig. S10).

Once the bipartite network has been constructed using a specific year's debt data, banks are phased-out either in a random order (the

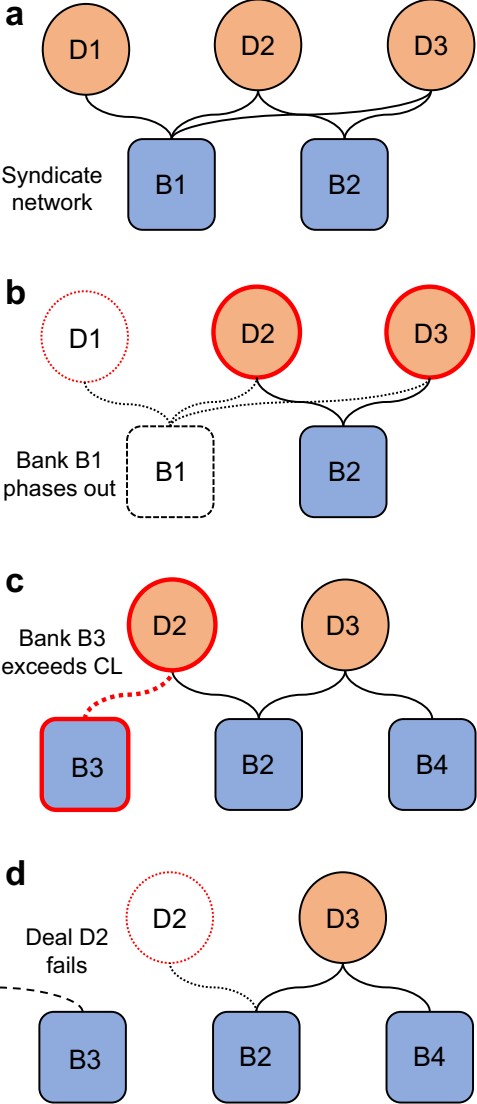

**Fig. 7 | Schematic for the fossil fuel phase-out model. a** Banks, B1 and B2, provide finance to deals, D1, D2, and D3 as a syndicate. **b** Bank B1 exits the fossil fuel sector, consequently, deals D1–D3 are impacted due to a shortfall in finance. If a project loses all of its financing, e.g., D1, it fails. **c** Deals D2 and D3 attempt to acquire substituted finance from banks who are not already in the syndicate, banks B3 and B4. However, bank B3 cannot finance deal D2 without breaching finance limits. **d** Without finance from bank B3, deal D2 collapses; note that bank B3 may remain active in other deals. Here, the phase-out efficiency is (D1+D2)/(D1+D2+D3), since deal D3 has survived due to substituted finance, but deals D1 and D2 have failed.

'random' scenario in Fig. 4), or in the order of the total value of fossil fuel deals they are involved in, with the largest bank first (the 'targeted' scenario in Fig. 4). In the regional scenario (Fig. 6), only the banks from a specific region are phased-out, with these banks phasing-out in a random order.

Starting with the first bank in the determined phase-out order, the node corresponding to that bank in the bipartite network is deleted. Each deal which previously had financing from the phased-out bank now has a shortfall in finance and requires a new banking partner in order to provide substitute finance. Deals which do not acquire substitute finance fail, and the node corresponding to that deal is removed from the bipartite network, including all edges from that deal to the syndicate partners who remain active in the sector (i.e., the banks which have not yet phased out).

We use two different methods to determine the candidate banks for providing substitute finance to deals-at-risk. In the first scenario—the "any substitute" scenario—we list all banks who are not already a syndicate partner in the deal-at-risk and have not phased-out from the fossil fuel sector. These banks are ordered at random and then, one by one, we check if the new bank can provide the substitute finance without exceeding its finance limit. If yes, that bank becomes a syndicate partner in the deal-at-risk, and that deal survives. If not, we proceed to the next candidate bank in the list. If all possible candidate banks are exhausted, the deal fails. In this scenario, historic syndicate relationships do not influence the banks chosen to provide substitute finance (although note that since existing syndicate partners in the deal-at-risk cannot be chosen to provide substitute capital, this element of the syndication network is retained).

In the second case—the "syndicate substitute" scenario—we list all banks who are not already a syndicate partner in the deal-at-risk and rank them according to the number of syndicates in which they were co-active with the banks currently active in the deal-at-risk's syndicate, with the bank co-active in the largest number of deals ranked first. Banks who are not co-active in any deals with the current syndicate are not ranked. Starting with a candidate bank from the top of the ranking, we check if the candidate can provide the substitute finance without exceeding its finance limit. If yes, that bank becomes a syndicate partner in the deal-at-risk, and that deal survives. If not, we proceed to the next candidate bank in the list. If all possible candidate banks are exhausted, the deal fails. In both scenarios, the process for finding substitute finance is repeated for all impacted deals until each deal affected by a bank phase-out has either acquired new finance or has failed. We then proceed to the next bank in the phase-out order and repeat the process of identifying impacted deals and searching for substitute finance. A simulation ends when all banks (or all banks from a specific region in the case of Fig. 6) have been phased out of the fossil fuel sector.

Two additional restrictions are tested using the syndicate substitute scenario, shown in Fig. 5. First, we limit the number of possible candidate banks for substitution by restricting the candidate list to the top N ranked banks, see Fig. 5b. Second, we consider a quasi-behavioural restriction in which we assert that a specific deal-at risk can only acquire substitute finance a fixed number of times before failing, thereby simulating a deal which has repeatedly lost financing and is therefore perceived as too risky. This is shown in Fig. 5c.

### Efficiency gap
The efficiency gap is defined as the difference between the total value of impacted deals after all banks have phased-out of the fossil fuel sector, relative to the total value of deals in the sector. Importantly, a single deal can count towards the impacted deals multiple times; once for each time a deal becomes 'at-risk' and successfully finds substitute finance, and once when a deal-at-risk fails. Hence, an efficiency gap of 0 means that impacted deals in the phase-out process always fail. Conversely, an efficiency gap larger than 0 means that deals are successfully substituting finance such that $1 of finance pulled from the sector leads to less than $1 of finance lost to fossil fuel deals. For simplicity, the efficiency gap is presented as a multiple of the total value of deals in the sector.

### Total relative efficiency
The appearance of an efficiency gap at the end of the phase-out process is associated with the inefficient withdrawal of finance throughout the phase-out process. We measure this using the 'total relative efficiency', defined as the ratio between the cumulative value of all failed deals up to the $n^{th}$ bank removal, normalised by the total value of all deals in the sector, and the cumulative value of all impacted deals up to the $n^{th}$ bank removal, normalised by the cumulative value of all impacted deals throughout the whole phase-out process.

### Phase-out multiplier
The 'phase-out multiplier' is defined as the ratio between the cumulative value of all failed deals up to the $n^{th}$ bank removal, divided by the cumulative value of the finance phased-out by the n banks that have chosen to directly exit the fossil fuel sector. Since syndication means that individual banks rarely fund a single fossil fuel deal by themselves, a deal failing due to the phase-out of finance by one bank can induce the phase-out of finance from syndicate partners if deals fail to acquire substitute finance. For example, if banks B1 and B2 are the syndicate partners in deal D1, each contributing half the deal's value in finance, then if bank B1 exits the sector and the deal fails to acquire substituted finance, then bank B2 is forced to also phase-out from deal D1. In this case, we have a phase-out multiplier of 2 since for each $1 phased-out of deal D1 by bank B1, $2 of total value has been phased out from the sector.

### Reporting summary
Further information on research design is available in the Nature Portfolio Reporting Summary linked to this article.

## Data availability
The data used in this study are available under restricted access as they remain the commercial property of Bloomberg. Licences to use these data can be obtained from Bloomberg who require compensation to cover the cost of producing and maintaining them. The data are then available for the term of the license.

## Code availability
Code used to simulate the phase-out model is available at https://github.com/LINKS-ERC/.

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

## Acknowledgements

J.R., S.K., F.L., and N.A. acknowledge funding from the European Research Council (ERC) under the European Union's Horizon 2020 research and innovation programme (grant agreement No. 802891). M.F. acknowledges the 100683EPID Project "Global Health Security Academic Research Coalition" SCH-00001-3391.

## Author contributions

J.R., M.F.: conceptualisation, methodology, analysis, writing–original draft. S.K., F.L., and M.G.: writing–review & editing. N.A.: conceptualisation, writing–review & editing, supervision.

## Competing interests

The authors declare no competing interests.
