## [Peer Review File · Nature Communications]

The challenge of phasing out fossil fuel finance in the banking sectorReviewers' Comments:

Reviewer #1:

Remarks to the Author:

I find the paper very innovative and interesting. This type of analysis of the syndicated loan network is, to my knowledge, unprecedented and its application to the financing of the fossil fuel sector is very relevant.

Still, I think the following points should be addressed before publication.

-Most importantly, I don't think it is appropriate to use what the authors define as 'finance limit percentage' as an analogue/synonym of capital requirements. I would strongly suggest to the authors to refrain to (mis-) use the notion of capital requirement altogether and to rather elaborate on potential ways to implement their finance limit in the context of green financial regulations (e.g. these could be linked to constraints on the carbon intensity of the portfolio for which there already exist reporting requirements).

- For the empirical analysis, it could also be interesting to look at the evolution of the number of deals in which banks act as arranger, agent or trustee, in order to assess whether banks have maintained/increased their fee-business related to FF although possibly decreasing their direct exposure.

-For the empirical analysis, it would be important to have an understanding of how the behavior in fossil fuel segment differs from the broader syndicated loan market. Is there anything specific to the fossil market or is it that the global syndicated loan business is changing ?

-In the model-based analysis, do the authors consider a link exists if there is joint participation in a syndicate for a fossil loan or for any type of loan ? The latter would seem to me a more appropriate indicator of the possibility of substitution.

-There are a number of issues with grammar. Please proofread.

Reviewer #2:

Remarks to the Author:

The paper provides a network analysis of the global bank-to-bank relationships created by syndicated loans to the fossil fuel sector between 2010 and 2021. The main findings of the paper are (i) that syndicated fossil fuel debt markets are resilient to phasing out scenarios and (ii) prudential regulations (in the form of capital requirements) have a key role in mediating the decline of funding flows towards the fossil fuel sector. The analysis is interesting, and the paper is well-written. I see the potential to spur interest in multiple scientific communities.

I provide my comments below, not in order of importance.

The presentation and the discussion of the data the authors use for the first part of the analysis are intriguing but can be improved. I have four comments/suggestions.

- First, it is not clear how the banks' exposition varies across different fossil fuel "sectors" (e.g. oil, coal, and gas), though phasing out certain fuels may be more relevant than others from a climate perspective. Does the data allow such a disaggregation? See also one of my other comments below.

- Second, I would suggest the authors be more specific in their claim that their "data are representative of the global banking sector". I wonder if and how the data cover debt relationships in economies that are relatively small in size but crucial in the fossil fuel sector, and especially the extraction side (e.g. Indonesia, Kuwait, Russia,...).

- Third, it would be important to account for the very nature of the credit relationships: public national banks providing loans (alone or in group) to a large national public oil company may be different from Goldman Sachs and JP Morgan providing credit to, say, ENI. Some more effort in this direction could uncover relevant elements for a better understanding of realistic and feasible phasing-out dynamics. Does the data the authors use contain this information?

- Fourth, I feel it would be relevant not just to look at the overall level of syndicated loans to fossil fuel companies but also (and perhaps more relevantly) to the share of such loans relative to the total liabilities of fossil-fuel firms. If their external funding through syndicated loans is easy to substitute with other instruments, reducing it may not as relevant as indirectly suggested by the analysis. If the contrary is true, the paper is even more valuable. I would appreciate a discussion.

As a second set of comments, I focus on the modeling exercise. The model is simple, data-driven, and insightful. I have four comments/suggestions.

- First, I believe the authors should improve the discussion - in the main text - of the two key features of the model: (i) how the "finance limit percentage" is set and (ii) how banks substitute one with the other in syndicated loans. In particular, the "finance limit percentage" is defined as a year-on-year change. Differently, capital requirements are typically defined as shares, in a way to limit the overall exposure to risks, rather than the variation in the exposure. I believe a discussion is needed here. Are results robust to a more standard "prudential policy"? Further, how banks are selected to replace one another in a loan is relevant: some more discussion and robustness in the main text may help to convince the reader about the robustness of the results. Indeed, the current rewiring rule is very simplistic. Moving some of the insights out the SI may be a starting point.

- Second, It may be interesting to show how effective the "finance limit percentage" is (Figure 3 and 4) relative to a more informative measure of phasing out, e.g. the share of banks phased out or the share of total loans phased out. Indeed, if the distribution of banks' size is fat-tailed (as it typically is) the total number of banks is not very informative of the dynamics in the network. This may also provide a stronger claim on the tipping point the authors find: indeed, the "tipping threshold" would

be expressed in a metric that I see as more policy-relevant (than the simple number of banks). Differently, yet related, is it sufficient to remove the, say, 4 largest providers of loans to the fossil fuel sector to reach the "tipping point"? I also have a brief request for clarification related to Figure 3: why does the 0% limit induce a phase-out multiplier that is decreasing in the number of banks?

- Third, the results the authors obtain are relevant beyond network models of transition risks. Indeed, they may be included in macroeconomic models with heterogeneous agents to assess the effects of phasing out funding to fossil fuel companies at the macroeconomic level. This link would be very appealing to policymakers and central bankers. A candidate model may be the DSK agent-based integrated assessment model, but there are others as well. I suggest the authors better account for such a potential impact of their work in the discussion.

- Fourth, the attempt to break down the results (e.g. at regional level) is interesting and may deserve more attention in the main text. The authors could show what happens if capital requirements are region-specific, all else being equal. Further, breaking down by fossil fuel (e.g. oil vs coal) may also be very relevant. I perfectly understand that many fossil fuel companies extract and refine more than a single "fuel", but I wonder whether loans' information the authors have access to allows going slightly more granular than the current manuscript does. At the very least, I feel there is some space in the main text to move part of the analysis which is now in the SI.

Finally, I suggest the authors to better connect their discussion on tipping points to the recent Global Tipping Points report.

Reviewer #3:

Remarks to the Author:

Many thanks to the authors. The research is super-interesting and relates to an area of finance that has not been well covered in the literature. I have more major comments below, which may reflect my understanding of the method not being completely clear. Some may be addressed with simple clarifications.

Major comments

1 In the available data, is there any way to differentiate between different fossil fuel types? For example, the US banks may have reduced lending only because of coal. At least in the descriptive parts of the paper, some disaggregation would be helpful.

2 Do we want fossil fuel lending to fall to zero as fast as possible? Although the point about existing fossil fuel assets exceeding Paris targets is well taken, do we know for sure that all lending is to bring new capacity on stream? If some lending is for other purposes, for example rolling over existing debt, or even low-carbon investments by fossil fuel companies, how would this appear in the dataset?

3 How exactly does the network linkages affect the simulations? I found this a bit difficult to understand, even after reading the methods. My interpretation is that when one bank is removed, its network linkages are also removed, and it can be replaced with any other banks within the remaining network. My first thought was that this makes the strength of individual network links redundant, which is a shame (although I think I understand there are some data limitations here anyway). It feels like more could be made of the network, e.g. favoring banks with 2+ linkages to the current syndicate. The sensitivity analysis starts to get to this, and it would better justify the space given to the earlier network analysis.

4 This comes to the core assumption of the network being fixed. Here some additional sensitivity would be good, e.g. about how cut-off banks might try to re-integrate themselves. For example, banks that were one-step removed could be more likely to join just by reaching out to existing contacts.

Minor comments

1 On line 121 there appears to be a contradiction between the importance of targeting large firms and the ease of substitution with other banks coming in described in line 66 and the rest of the paragraph. I get the point that it might be different for large banks, but this could be clearer.

2 The sensitivity results from removing large and central banks makes sense and are not surprising – is it possible to confirm it is the network centrality rather than size that gives the result?

3 One missing reference is Gregor Semieniuk's paper in Nature Climate Change. I see one of Stefano Battiston's work is referenced – I thought there were more by him and Irene Monasterolo.

4 I appreciate that there is a discussion of regional policies later on, but I would mention this when first introducing the random phase-out in line 277, to help the reader with the obvious question.

RESPONSE TO REVIEWERS

Reviewer #1 (Remarks to the Author):

I find the paper very innovative and interesting. This type of analysis of the syndicated loan network is, to my knowledge, unprecedented and its application to the financing of the fossil fuel sector is very relevant.

We thank the reviewer for this positive appraisal of our work.

Still, I think the following points should be addressed before publication.

-Most importantly, I don't think it is appropriate to use what the authors define as 'finance limit percentage' as an analogue/synonym of capital requirements. I would strongly suggest to the authors to refrain to (mis-) use the notion of capital requirement altogether and to rather elaborate on potential ways to implement their finance limit in the context of green financial regulations (e.g. these could be linked to constraints on the carbon intensity of the portfolio for which there already exist reporting requirements).

We thank the reviewer for this point and agree that our use of terminology was imprecise with respect to capital requirements rules. We have now altered our text such that the 'finance limit percentage' refers generically to any instrument of prudential regulation that restricts banks holdings and activities in high-risk sectors and we do not link our modelling specifically to capital requirements rules.

- For the empirical analysis, it could also be interesting to look at the evolution of the number of deals in which banks act as arranger, agent or trustee, in order to assess whether banks have maintained/increased their fee-business related to FF although possibly decreasing their direct exposure.

We thank the reviewer for this suggestion that would indeed strengthen our results relating to banks shifting their activity away from direct lending to fee-business. Unfortunately, we do not have information in our data on the particular role banks undertake within a syndicate, so this is not possible to do at present. We have removed any speculation in our discussion regarding fee-related business to alleviate any queries in the mind of the reader. Should the data become available, providing evidence on this point would be highly valuable.

- For the empirical analysis, it would be important to have an understanding of how the behaviour in fossil fuel segment differs from the broader syndicated loan market. Is

there anything specific to the fossil market or is it that the global syndicated loan business is changing ?

We thank the reviewer for this suggestion and agree that comparison between the fossil fuel and global syndicated loan market is important context for our findings. We have retrieved data from Bloomberg on the global syndicated loan market and added a discussion to our main text and a supplementary figure:

“Dynamics in the syndicated fossil fuel debt market are distinct from the global syndicated loan market as it experiences sector-specific phases of growth and contraction related to economic, geopolitical, and technological factors (Fig. S1).”

Our additional supplementary figure shows, for example, that the syndicated fossil fuel debt market did not experience the same post-Covid recovery as observed in the global syndicated debt market, although it remains the primary source of fundraising for fossil fuel companies. We can therefore have confidence that our results are specific to the fossil fuel sector.

-In the model-based analysis, do the authors consider a link exists if there is joint participation in a syndicate for a fossil loan or for any type of loan ? The latter would seem to me a more appropriate indicator of the possibility of substitution.

Thank you for this comment. In the current model a link exists between two banks if they both participate in the same fossil fuel loan syndicate, not syndicates for any type of loan. While we agree that the latter would be interesting as a robustness check, we unfortunately do not have syndication data for other loan types. We would not expect that using syndicate relationships from any loan type, as opposed to fossil fuel syndicates only, will substantially affect our results given that the network of lending relationships is very dense with respect to the major banks. The fossil fuel sector has a long track record of obtaining sizable investments from financial markets, around \$1 trillion annually over the last 8 years (IEA), with all major financial institutions and market funds holding exposure to these assets. Moreover, we believe that by using sector-specific syndicated fossil fuel loan data we capture bank behaviour at the sector level, which is distinct from the global-level syndicated debt (see discussion above), and thus reflects sector-specific relationships and dynamics. In the revised manuscript, we have added new text to minimise any ambiguity and make clear the assumptions which go into determining the structure of the syndicate network.

“The structure of the syndicated lending network considers banks which are syndicate partners in fossil fuel deals only, and it does not consider banking relationships in other sectors.”

We have also added new model variants to the main text (which were previously in the supplement) to demonstrate how related model features affect the phase-out process e.g., candidate banks are chosen for substitution randomly or based on fossil fuel lending relationships.

-There are a number of issues with grammar. Please proofread.

Thank you for this comment. We have thoroughly checked the text and have fixed any grammar issues identified.

Reviewer #2 (Remarks to the Author):

The paper provides a network analysis of the global bank-to-bank relationships created by syndicated loans to the fossil fuel sector between 2010 and 2021. The main findings of the paper are (i) that syndicated fossil fuel debt markets are resilient to phasing out scenarios and (ii) prudential regulations (in the form of capital requirements) have a key role in mediating the decline of funding flows towards the fossil fuel sector. The analysis is interesting, and the paper is well-written. I see the potential to spur interest in multiple scientific communities.

We thank the reviewer for this positive appraisal of our work.

I provide my comments below, not in order of importance.

The presentation and the discussion of the data the authors use for the first part of the analysis are intriguing but can be improved. I have four comments/suggestions.

- First, it is not clear how the banks' exposition varies across different fossil fuel "sectors" (e.g. oil, coal, and gas), though phasing out certain fuels may be more relevant than others from a climate perspective. Does the data allow such a disaggregation? See also one of my other comments below.

We thank the reviewer for this comment. Certainly, this dimension of fossil fuel financing is relevant for planning a just and managed transition at country-level and would thus have implications for the phasing-out of fossil fuel finance. However, our data does not

allow us to identify whether debt is being used to finance coal, oil, or gas activities as we have no information on borrowers. We therefore focus our modelling on the supply side of fossil fuel debt financing, in particular tipping point dynamics within syndicated lending networks and the role of prudential regulation. We have updated the description of our data and discussion of model limitations to make this clearer:

“Bloomberg data provides industry-leading coverage of the global banking sector, with reported global debt volumes surpassing those of alternative financial data providers by 29% (see SI Methods). Nonetheless, the data lacks information on fossil fuel borrowers, such as their location or asset type (coal, oil, or gas). While these dimensions of the fossil finance landscape are relevant for just transition planning at country-level, we focus our analysis on the dynamics of global fossil fuel debt supply, in support of the efforts of international financial supervisors.”

- Second, I would suggest the authors be more specific in their claim that their “data are representative of the global banking sector”. I wonder if and how the data cover debt relationships in economies that are relatively small in size but crucial in the fossil fuel sector, and especially the extraction side (e.g. Indonesia, Kuwait, Russia,...).

We thank the reviewer for this suggestion and have endeavoured to improve and clarify our description of the data and its coverage. We now compare the volumes of debt in our dataset from Bloomberg with data from alternative industry-leading providers of financial data: Refinitiv and Thomson Reuters. Reported debt volumes are 29% higher in Bloomberg than from independent estimates, affirming that Bloomberg has excellent coverage of global bank debt. The Bloomberg data does not report the borrowing firms so we cannot disaggregate the data into fossil fuel debt by country. However, we can have confidence that the Bloomberg data represents financing of the major fossil fuel economies, given its excellent coverage.

In addition, our analysis and modelling of the global banking sector focuses on debt supply rather than demand (see response above). It captures a large share of the global banking sector activity in financing fossil fuels as well as the key financial institutions and thus the macro-dynamics of the sector.

Furthermore, in our new manuscript we have added in our main text new regional phase-out scenarios to highlight the impact of regional lending relationships on fossil finance phase-out dynamics that may be relevant for domestic regulation at country level.

- Third, it would be important to account for the very nature of the credit relationships: public national banks providing loans (alone or in group) to a large national public oil company may be different from Goldman Sachs and JP Morgan providing credit to, say, ENI. Some more effort in this direction could uncover relevant elements for a better understanding of realistic and feasible phasing-out dynamics. Does the data the authors use contain this information?

We thank the reviewer for this comment and agree that this is an interesting and important dimension of the fossil fuel financing landscape. Unfortunately, as previously mentioned, we do not have information on the borrowing firms and so an analysis of bank-borrower relationships is not possible. However, we conducted additional analysis to ascertain the portion of global bank debt provided by commercial and public banks in our data and confirmed that the majority (96%) was provided by commercial banks over the period analysed. This additional information is now included in the supplementary information. While important, public banks therefore represent a small part of the global fossil fuel financing landscape and the particular credit relationship they represent is therefore not central to our analysis. What's more, regulation of public finance institutions is a distinct policy question that is not within the scope of global financial regulation e.g., the Basel Framework enacted by the BCBS, that motivate our analysis.

Fourth, I feel it would be relevant not just to look at the overall level of syndicated loans to fossil fuel companies but also (and perhaps more relevantly) to the share of such loans relative to the total liabilities of fossil-fuel firms. If their external funding through syndicated loans is easy to substitute with other instruments, reducing it may not as relevant as indirectly suggested by the analysis. If the contrary is true, the paper is even more valuable. I would appreciate a discussion.

We thank the reviewer for this comment and agree that a discussion of these points is highly relevant. We have clarified these points in our introduction:

“Debt syndication is an important facility for the capital-intensive fossil fuel sector as it allows deals to be made that are too large for the balance sheets of an individual bank and spreads risk among syndicate participants. In this way, syndication amplifies the impact of individual banks’ investments and drives significant financial support to the fossil fuel industry. Syndicated loans accounted for 66% of global fossil fuel finance in 2018, followed by bonds (29%) and equity instruments (5%)⁶. Indeed, bank loans may become ever more critical to fossil fuel firms in the future as capital markets are pricing climate-related financial risk^{14, 15}, limiting the possibility for bank loans to be substituted with market-based finance¹⁶.”

As a second set of comments, I focus on the modeling exercise. The model is simple, data-driven, and insightful. I have four comments/suggestions.

- First, I believe the authors should improve the discussion - in the main text – of the two key features of the model: (i) how the “finance limit percentage” is set and (ii) how banks substitute one with the other in syndicated loans. In particular, the “finance limit percentage” is defined as a year-on-year change. Differently, capital requirements are typically defined as shares, in a way to limit the overall exposure to risks, rather than the variation in the exposure. I believe a discussion is needed here. Are results robust to a more standard “prudential policy”? Further, how banks are selected to replace one another in a loan is relevant: some more discussion and robustness in the main text may help to convince the reader about the robustness of the results. Indeed, the current rewiring rule is very simplistic. Moving some of the insights out the SI may be a starting point.

Thank you for these useful comments. Firstly, we agree with the reviewer that our finance limit percentage is not directly analogous to capital requirements rules which are typically defined as shares of a banks’ overall exposure and our terminology was misleading. In the revised manuscript we now discuss prudential policy more generically and the finance limit percentage is described as simply a modelling choice to implement a cap on banks’ fossil fuel activity, motivated by empirical evidence and data availability, discussed further below. The revised text also discusses how the finance limit percentage is set in more detail:

“We model this cap by defining the ‘finance limit percentage’ -- the maximum annual percentage increase in a banks’ fossil fuel lending (see Methods) – which serves as a generic analogue for prudential instruments that restrict banks holdings and activities in high-risk sectors. For example, if Bank A held fossil fuel assets worth \$1bn in 2021, a 10% finance limit percentage would set a cap on Bank A’s 2022 fossil fuel assets at \$1.1bn. ”

Taking on board your comments, we have added a new model variant to the supplement (Fig. S4), where instead of capping substitution using the finance limit percentage, we set a cap on banks’ absolute exposure to the fossil fuel sector (percentage of total assets held) in line with more standard prudential policy. Our model is robust to this change. This model variant shows qualitatively equivalent behaviour to the model variant using the finance limit percentage – the stricter the cap on exposure, the faster the model reaches the tipping point from inefficient to efficient phase-out. However, in practice only very few banks are significantly exposed to the sector (see Fig. 1b). An exposure limit of, for example, 1% would be effective at forcing some major

banks to phase-out from the sector (Fig. S4), but this limit would have no effect on the large number of banks who are currently far below this level of exposure. We have added this discussion to the main text:

“Model results are qualitatively robust to setting the cap as a limit to fossil fuel exposure (e.g., lending cannot exceed 1% of total assets, Fig. S4). However, given significant variability in exposure across the largest banks (Fig. 1b), such flat-rate limits only affect the most exposed banks (notably in Canada and the USA) unless the exposure limit is very strict.”

There are also several other important caveats to this model variant which are grounds for its relegation to the supplement. Firstly, due to data availability, we only have reliable exposure values for the largest thirty banks in our dataset. For the remaining banks we model their exposure using a gamma distribution following the observed distribution from the banks for which we have data. Secondly, for banks which do fall below the exposure limit, there are potentially other behavioural and strategic constraints which might prevent them from increasing their fossil fuel investments and providing substitute finance. However, these factors are complex and cannot realistically be incorporated into our model. As we stress in the manuscript, the primary motivation of our work is to encourage regulators to carry out more detailed analyses of capital substitution using the prudential data that we do not have access to. This new model variant and these important caveats are now discussed in detail in the revised manuscript and supplement.

We are also happy to improve the discussion of how banks are substituted in the main text to remove any ambiguities, and significant efforts have been made to expand the methods where we discuss the model rules. We have also added Figure 5 to the revised text which compares two implementations of the substitution rule: (i) any active bank, independent of the structure of the syndicate network, may act as a candidate for capital substitution if they do not exceed their finance limit percentage – referred to in the text as the “any substitute” model, (ii) only banks who are co-active with at least one other bank in the current deal-at-risk’s syndicate are candidates for substitution – referred to in the text as the “syndicate substitute” model. These candidates are ranked according to the number of co-active deals with the current bank, with the most active bank chosen as the first candidate for substitution. If this bank cannot provide substitute finance, we proceed to the second bank in the ranking, and so on. These results were previously shown in the supplement but have now been moved to the main text where they are discussed in detail. We have also added a new scenario discussing regional regulations which highlights the importance of the syndicated network structure in

determining how effectively prudential regulations can lead to the phase-out of fossil fuel finance from a specific region, see the new Figure 6.

- Second, It may be interesting to show how effective the “finance limit percentage” is (Figure 3 and 4) relative to a more informative measure of phasing out, e.g. the share of banks phased out or the share of total loans phased out. Indeed, if the distribution of banks’ size is fat-tailed (as it typically is) the total number of banks is not very informative of the dynamics in the network. This may also provide a stronger claim on the tipping point the authors find: indeed, the “tipping threshold” would be expressed in a metric that I see as more policy-relevant (than the simple number of banks). Differently, yet related, is it sufficient to remove the, say, 4 largest providers of loans to the fossil fuel sector to reach the "tipping point"? I also have a brief request for clarification related to Figure 3: why does the 0% limit induce a phase-out multiplier that is decreasing in the number of banks?

Thank you for these important points. Firstly, we are happy to adjust the figures to show the share of banks which have phased out as opposed to the total number of banks. This change has been made for Figures 3, 4 and 5. We have also added the new panel b in Figure 4 where, instead of plotting the phase out efficiency as a function of the share of banks which have phased out, we plot the efficiency as a function of the finance phased-out by the banks exiting the sector, thereby weighting banks according to the volume of their fossil fuel assets. In doing so, we show that random phase-out (banks exit in a random order) and targeted phase-out (banks exit in order of the volume of their fossil fuel investments) are similar in terms of the total finance which must be phased-out in order to reach the tipping point but there is an important increase in phase-out efficiency in the targeted phase-out scenario:

“Although the efficiency tipping point occurs at comparable values in both scenarios in terms of finance phased out (Fig. 4b), the targeted scenario does result in an important increase in phase-out efficiency, presented as a decrease in the efficiency gap (Fig. 4c). This is because, once the largest banks have phased-out in the targeted scenario, unlike in the random scenario, the remaining banks are individually too small to provide substitute finance to deals-at-risk, even if there is sufficient finance available across the sector as a whole.”

Secondly, the reviewer is correct that it is sufficient to remove the largest banks from the fossil fuel sector in order to reach the tipping point, rather than phasing out banks at random – the exact number of major banks which need to phase out to reach the tipping point depends on the finance limit percentage. This was shown in the original manuscript as Figure 4 and corresponds to the difference between the random and

targeted phase-out scenarios. We have revised the text to further emphasise this point (see above).

Finally, thank you for highlighting the lack of clarity regarding the phase-out multiplier. To understand why the multiplier reduces over the phase-out period with a 0% limit, it is important to realise that the 0% limit does not allow for any substitution. In this scenario, if a deal loses a single syndicate partner, no other bank can substitute that capital and all affected deals must fail. To understand why the multiplier reduces throughout the phase-out process when the limit is set to 0%, notice that the model has already passed the tipping point before the phase-out process starts. This means that the direct phase-out of finance immediately induces the indirect phase-out of capital (multiplier > 1), and the ability to induce indirect phase-out is the strongest when the network of syndicate connections between banks is the densest at the start of the phase-out process. As banks sequentially phase-out, the syndicate network becomes sparser which is why the multiplier gradually decreases over time but remains larger than zero. In the revised manuscript, we have added new text to explain this phenomenon:

“Figure 3c shows the ‘phase-out multiplier’, which is the ratio of the finance removed directly by a phasing-out bank to the value of deals failing due to its phase-out (see Methods). When substitution is possible, the phase-out multiplier is zero before reaching the tipping point, since all phased-out finance can be replaced. Beyond the tipping point, two types of substitution dynamics emerge. With sufficiently large finance limit percentages (approximately $>100\%$) the phase-out multiplier never exceeds 1. This means that, on average, \$1 removed by the phased-out bank results in less than \$1 of finance being phased-out from the sector. Conversely, with sufficiently small finance limit percentages (approximately $<100\%$), the multiplier exceeds 1. This indicates that \$1 removed by the phasing-out bank results in more than \$1 of finance lost to the sector. This occurs because the direct phase-out of an individual bank induces indirect phase-out of its syndicate partners as their finance is withdrawn from failed deals, while they remain active in the sector. Finally, when the finance limit percentage is 0%, no substitution is possible, and phase-out efficiency remains at 1 throughout the phase-out process. In this case, every phasing-out of a bank indirectly induces phase-out of other banks, resulting in a large phase-out multiplier. This effect is strongest at the start of the phase-out process when the average number of deals per active bank is highest (since a single bank’s phase-out results in many failed deals). As the phase-out continues, the average number of deals per active bank decreases (as many deals have already failed), and the indirect phase-out effect weakens, leading to gradually decreasing phase-out multiplier.”

- Third, the results the authors obtain are relevant beyond network models of transition risks. Indeed, they may be included in macroeconomic models with heterogeneous agents to assess the effects of phasing out funding to fossil fuel companies at the macroeconomic level. This link would be very appealing to policymakers and central bankers. A candidate model may be the DSK agent-based integrated assessment model, but there are others as well. I suggest the authors better account for such a potential impact of their work in the discussion.

We thank the reviewer for pointing out this important application of our results. Indeed, an agent-based integrated assessment model would be a highly appropriate and relevant framework within which to incorporate our detailed representation of bank lending. We have included this line in our discussion:

“Furthermore, the networks dynamics identified here could be used to develop more detailed, bottom-up representations of the banking sector to be used in agent-based integrated assessment models assessing climate damages (Lamperti et al., Ecological Economics, 2018).”

- Fourth, the attempt to break down the results (e.g., at regional level) is interesting and may deserve more attention in the main text. The authors could show what happens if capital requirements are region-specific, all else being equal. Further, breaking down by fossil fuel (e.g. oil vs coal) may also be very relevant. I perfectly understand that many fossil fuel companies extract and refine more than a single “fuel”, but I wonder whether loans’ information the authors have access to allows going slightly more granular than the current manuscript does. At the very least, I feel there is some space in the main text to move part of the analysis which is now in the SI.

Thank you for these useful comments. In the revised manuscript, we have made a substantial effort to improve our discussion of regional scenarios. Specifically, we now model the efficacy of finance phase-out if only banks from a specific region exit the fossil fuel sector and map the interdependence between banks from different regions to reveal how external finance may prevent the successful phase-out of finance. These results are shown in the new Figure 6 and are discussed in detail on pages 15-16.

Regarding the breakdown of deals according to fossil fuel type (oil vs coal), we agree that this would be very interesting. However, unfortunately our data is not sufficiently granular to allow for this analysis. In the revised manuscript, we state this limitation and also note that this analysis would be interesting in future work.

Finally, we have moved the results for different implementations of the substitution rule, and additional restrictions on the number of candidate banks for substitution and the number of times a deal can receive substitute finance, to the main text from the SI, see Figure 5.

Finally, I suggest the authors to better connect their discussion on tipping points to the recent Global Tipping Points report.

Thank you, this is a very important connection to make and inspiration for this work - we have added it to our discussion:

“...Indeed, tipping point dynamics in the banking sector have the potential to spill over into other sectors leading to a cascade of nonlinear changes that could drive transformative shifts in the real economy (Lenton et al., The Global Tipping Points Report 2023).”

Reviewer #3 (Remarks to the Author):

Many thanks to the authors. The research is super-interesting and relates to an area of finance that has not been well covered in the literature. I have more major comments below, which may reflect my understanding of the method not being completely clear. Some may be addressed with simple clarifications.

We thank the reviewer for this positive appraisal of our work.

Major comments

1 In the available data, is there any way to differentiate between different fossil fuel types? For example, the US banks may have reduced lending only because of coal. At least in the descriptive parts of the paper, some disaggregation would be helpful.

We thank the reviewer for this comment. Certainly, this dimension of fossil fuel financing is relevant for planning a just and managed transition at country-level and would thus have implications for the phasing-out of fossil fuel finance. However, our data unfortunately does not allow us to identify whether debt is being used to finance coal, oil, or gas activities as we have no information on borrowers. We therefore focus our modelling on the supply side of fossil fuel debt, in particular tipping point dynamics within syndicated lending networks and the role of prudential regulation. We have

updated the description of our data and discussion of model limitations to make this clearer:

“Bloomberg data provides industry-leading coverage of the global banking sector, with reported global debt volumes surpassing those of alternative financial data providers by 29% (see SI Methods). Nonetheless, the data lacks information on fossil fuel borrowers, such as their location or asset type (coal, oil, or gas). While these dimensions of the fossil finance landscape are relevant for just transition planning at country-level, we focus our analysis on the dynamics of global fossil fuel debt supply, in support of the efforts of international financial supervisors.”

2 Do we want fossil fuel lending to fall to zero as fast as possible?

-Although the point about existing fossil fuel assets exceeding Paris targets is well taken, do we know for sure that all lending is to bring new capacity on stream? If some lending is for other purposes, for example rolling over existing debt, or even low-carbon investments by fossil fuel companies, how would this appear in the dataset?

We thank the reviewer for this comment. We do not have information on whether the loans in our dataset are rolling over existing debt or financing new capacity and it is likely that both cases are present in the data. Low-carbon investments by fossil fuel firms should not be present in the data as these would be labelled as green/sustainable loans by Bloomberg. Ultimately, both the financing of new and old capacity must be phased-out for the transition, so both sources of finance are relevant for Paris targets. We are pleased to add nuance with respect to this point in the introduction:

“Projected emissions from existing fossil fuel assets could push the world past 2°C of warming within decades^{4, 5}. This means investment in new fossil fuel assets is incompatible with agreed climate targets, and the financing of existing assets must be phased-out in line with well-defined decarbonisation pathways for a just and orderly transition. As banks are the primary source of financing for fossil fuel companies⁶, their role in sustaining the ‘business-as-usual’ fossil fuel economy is now central to the wider climate debate⁷⁻⁹. ”

The timing of a fossil fuel finance phase-out is a highly salient issue with respect to climate justice and so we always refer in our manuscript to a ‘timely’, ‘well-managed’ and ‘just’ transition rather than a ‘fast as possible’ transition. However, our analysis explores the transition in terms of bank substitution dynamics and not timing in a

quantitative sense. This is a critical avenue for future research, and we have emphasised this point further in the introduction:

“In isolating just one module of the finance ecosystem - namely the network of syndicated lending relationships which determines who lends with whom - our study is distinct from the important stream of literature on financial stability which describes shock transmission through networks of interbank liabilities²⁹⁻³⁴. Our results could feed into the related stress-testing tools and macroeconomic models³⁵⁻³⁹ to provide a more detailed and realistic representation of the banking sector and its transition dynamics which will be necessary to coordinate a managed transition.”

3 How exactly does the network linkages affect the simulations? I found this a bit difficult to understand, even after reading the methods. My interpretation is that when one bank is removed, its network linkages are also removed, and it can be replaced with any other banks within the remaining network. My first thought was that this makes the strength of individual network links redundant, which is a shame (although I think I understand there are some data limitations here anyway). It feels like more could be made of the network, e.g., favouring banks with 2+ linkages to the current syndicate. The sensitivity analysis starts to get to this, and it would better justify the space given to the earlier network analysis.

We thank the reviewer for raising this ambiguity. In the revised manuscript we have made a major effort to improve the discussion of the model rules and remove any ambiguity in how the model is defined. Specifically, we have updated the methods, clarified the discussion of the model in the main text, and have included a comparison of model variants in the main text rather than in the supplement (see the new Figure 5).

We previously tested two versions of the model: (1) Any bank, independent of network links, may act as a candidate for substitution (now referred to as the ‘any substitute’ scenario) - this was previously shown in the main text. (2) Only banks with existing relationships to the current syndicate are candidates for substitution (now referred to as the ‘syndicate substitute’ scenario) - this was previously in the supplement and has now been moved to the main text. In line with the reviewer’s suggestion, we have also added a variant of the syndicate substitute model in which candidate banks are ranked in order of the number of deals in which they are co-active with the current syndicate, and we consider additional model variants where we limit substitution to only the banks with the strongest relationships to the current syndicate (see Fig 5).

These results, and the different model implementations, are now discussed in detail in the revised manuscript.

4 This comes to the core assumption of the network being fixed. Here some additional sensitivity would be good, e.g. about how cut-off banks might try to re-integrate themselves. For example, banks that were one-step removed could be more likely to join just by reaching out to existing contacts.

We thank the reviewer for this comment. While we agree that this is potentially interesting analysis, in our opinion it primarily relates to behavioural decisions made by banks which are very hard to accurately include in our model. The various model variants now discussed in our manuscript do consider different methods for determining the candidates for finance substitution such as using the strength of a bank's connections with candidate banks and do incorporate very simple behavioural limits which relate to deals being perceived as too risky if they repeatedly lose finance (see Fig 5c). However, in our view modelling how banks may re-enter the market is too complex a feature to include in a simple model such as ours. As we note in the main text, our model is intended to be illustrative and spur further work - by central banks and others - in the future, where these additional behavioural factors may be considered.

Minor comments

1 On line 121 there appears to be a contradiction between the importance of targeting large firms and the ease of substitution with other banks coming in described in line 66 and the rest of the paragraph. I get the point that it might be different for large banks, but this could be clearer.

Thank you for pointing this out, we have adjusted these statements to remove any confusion.

Line 66: *This has been reported in Australia's coal lending market, for example, as Australian banks have pulled out of large loan syndicates to be replaced by Chinese and Japanese lenders²².*

Line 121: *Phasing-out fossil finance provided by the top tier of lenders would significantly limit the amount of finance available for substitution, due to their size and market dominance, and their activity could thus be decisive in setting the pace of the phase-out.*

2 The sensitivity results from removing large and central banks makes sense and are not surprising – is it possible to confirm it is the network centrality rather than size that gives the result?

Thank you for this important comment. This relates to whether or not the candidate banks for finance substitution are based on the syndicate network or not. Previously, we compared two model variants to investigate this; one where network structure (and thus centrality) is ignored (the any substitute rule), and one where candidate banks for substitute capital are determined by the structure of the syndicate network (syndicate substitute rule). The former was discussed in the main text, and the latter in the supplement, which we acknowledge led to a lack of clarity. In the revised manuscript we compare the two model variants in Figure 5a to explicitly demonstrate the importance of network structure on syndication. Note, however, that bank size does play a major role in substitution dynamics as shown in Figure 4b.

The centrality of banks in the syndicate network is most important when considering regional phase-out scenarios. In the revised manuscript we have added a new regional scenario which discusses whether regional prudential regulations can enable the effective phase-out of capital from a specific region. This is shown in the new Figure 6 and is discussed on pages 15-16. As part of this discussion, we note how regional interdependence is directly related to the centrality of banks from specific regions:

“...This reveals several important relationships between regions. Firstly, when non-US regions exit the sector, US banks are the most likely candidates to provide substitute finance (indicated by the arrows pointing towards the US node) illustrating their global importance. Second, US-banks have strong, mutually supporting relationships with Japanese banks and Canadian banks. In contrast, while US, Japanese and Canadian banks are likely to substitute finance phased-out by EU banks (arrows from the EU to these countries), EU banks unlikely to provide substitute finance if Canadian and Japanese banks phase-out (no arrows from Japan or Canada to EU), and EU banks are of only minor importance to US-banks. This is consistent with the observations in Figure 2 of the growing centrality of Japanese banks, and the receding centrality of EU banks. Finally, UK and Chinese banks only have meaningful relationships with US-based banks, but in the latter case the total size of capital flows is small.”

3 One missing reference is Gregor Semieniuk’s paper in Nature Climate Change. I see one of Stefano Battiston’s work is referenced – I thought there were more by him and Irene Monasterolo.

We thank the reviewer for pointing out this key literature. We have now added three additional references to our text:

- *Semieniuk, G., et al., Stranded fossil-fuel assets translate to major losses for investors in advanced economies. Nature Climate Change, 2022. 12(6): p. 532-538.*
- *Battiston, S., et al., Accounting for finance is key for climate mitigation pathways. Science, 2021. 372(6545): p. 918-920.*
- *Battiston, S., Y. Dafermos, and I. Monasterolo, Climate risks and financial stability. 2021, Elsevier. p. 100867.*

4 I appreciate that there is a discussion of regional policies later on, but I would mention this when first introducing the random phase-out in line 277, to help the reader with the obvious question.

Thank you for this comment. We have revised the text to highlight the regional scenarios earlier in the paper. We have also expanded our discussion of the regional scenarios to better illustrate the regional interdependence between banks.

Reviewers' Comments:

Reviewer #1:

Remarks to the Author:

I am satisfied with the revision.

Reviewer #2:

Remarks to the Author:

I would like to thank the authors for their revision and replies. The paper is very interesting, has significantly improved and I recommend its publication.

Reviewer #3:

Remarks to the Author:

Many thanks to the authors for responding to my comments and suggestions. I am happy that my concerns have been addressed.

The only issue I would like to raise at this stage was I found Figure 1c quite difficult to follow. Either because it is the thicker/colored bars or this is how the ranking is done, I intuitively tried to match these to the top axis. It took a couple of minutes to realize that these were linked to the lower x axis - and only then I figured out how the axis color was meant to be an indication.

One other minor clarification on Figure 1a. The values are in nominal terms, right? If so the claim on line 132 that bank lending increased by 8% could be a bit misleading, because in real terms lending would have slightly declined?

RESPONSE TO REVIEWERS

Reviewer #3

Many thanks to the authors for responding to my comments and suggestions. I am happy that my concerns have been addressed.

The only issue I would like to raise at this stage was I found Figure 1c quite difficult to follow. Either because it is the thicker/colored bars or this is how the ranking is done, I intuitively tried to match these to the top axis. It took a couple of minutes to realize that these were linked to the lower x axis - and only then I figured out how the axis color was meant to be an indication.

One other minor clarification on Figure 1a. The values are in nominal terms, right? If so the claim on line 132 that bank lending increased by 8% could be a bit misleading, because in real terms lending would have slightly declined?

We thank the reviewer for their final comments to improve our manuscript and we have made the following amendments:

- i) In figure 1c the labelling and colouring of the axes have been inverted so the eye is more naturally drawn to the correct axes i.e. the top x-axis now corresponds to the thick coloured bars as the reviewer anticipated.
- ii) The wording in line 132 has been changed to avoid any misleading interpretation with respect to nominal vs. real values:

'Banks provided \$592 billion of bonds and loans for oil, gas, and coal companies in 2021, compared to a yearly average of \$584 billion between 2010 and 2016 (the year the Paris Agreement was signed).'